# Constant Bit-size Transformers Are Turing Complete

**Qian Li**[*]
Shenzhen International Center For Industrial And Applied Mathematics,
Shenzhen Research Institute of Big Data
Shenzhen, China
`liqian.ict@gmail.com`

**Yuyi Wang**[*]
CRRC Zhuzhou Institute & Tengen Intelligence Institute
Zhuzhou, China
`yuyiwang920@gmail.com`

## Abstract

We prove that any Turing machine running on inputs of arbitrary length can be simulated by a constant bit-size transformer, as long as the context window is sufficiently long. This improves previous works, which require scaling up either the model's precision or the number of parameters on longer inputs. Furthermore, we prove that the complexity class $\mathsf{SPACE}[s(n)]$ exactly characterizes the expressive power of a constant bit-size transformer with a context window of length $s(n)$. Our approach relies on simulating Post machines, a Turing-complete computational model. Post machines can be modeled as automata equipped with a queue, exhibiting computational behaviors naturally aligned with those of transformers. The behavioral similarity between transformers and Post machines may offer new insights into the mechanisms underlying the reasoning abilities of transformers.

## 1 Introduction

Transformer-based large language models (LLMs) with chain-of-thought (CoT) steps [Achiam et al., 2023, Gemini et al., 2023, Anthropic, 2024, Dubey et al., 2024, Guo et al., 2025] have demonstrated exceptional capabilities across a wide range of complex reasoning tasks, such as mathematical problem solving and code generation [Hendrycks et al., 2021, Jimenez et al., 2023, Rein et al., 2023]. These remarkable empirical successes raise a fundamental question: What enables transformers to support such general reasoning capabilities, and what are the limits of this capability? Understanding this question is not only of theoretical interest but also of practical significance for further enhancing the reasoning abilities of transformers.

Motivated by this fundamental question, a line of theoretical works [Pérez et al., 2021, Bhattamishra et al., 2020, Merrill and Sabharwal, 2023, 2024, Li et al., 2024, Qiu et al., 2024, Zubić et al., 2024, Merrill and Sabharwal, 2025, Yang et al., 2025] have studied the reasoning abilities of transformers through the lens of expressiveness. A central result from these works is that transformers (using CoT steps) are Turing complete; that is, transformers have sufficient expressive power to simulate arbitrary Turing machines (TMs). However, in all these results, the bit-size of the transformers (i.e., the total number of bits representing the model) is not constant: either the precision bit [Pérez et al., 2021, Bhattamishra et al., 2020, Merrill and Sabharwal, 2024, Qiu et al., 2024] or the embedding dimension [Li et al., 2024] must grow with the input length, meaning that larger models are required to handle longer inputs. This scaling requirement raises a fundamental barrier to approaching transformer-based

---

[*]The authors are listed in alphabetical order.

39th Conference on Neural Information Processing Systems (NeurIPS 2025).

Table 1: Comparing the existing Turing-completeness proofs in terms of required precision, embedding dimension, effective window size, and CoT length. We focus on the nontrivial case where $t(n), s(n) \geq n$. We remark that these constructions typically do not employ the vanilla architecture but instead introduce specific modifications; see Appendix A for a summary.

| Source | Precision | Dimension | Window | CoT per TM-step |
|---|---|---|---|---|
| Pérez et al. [2021] | $O(\log t(n))$ | $O(1)$ | $n + t(n)$ | 1 |
| Bhattamishra et al. [2020] | unbounded | $O(1)$ | $n + t(n)$ | 1 |
| Merrill and Sabharwal [2024] | $O(\log t(n))$ | $O(1)$ | $n + t(n)$ | 1 |
| Li et al. [2024]* | $O(1)$ | $O(\log t(n))$ | $O(t(n) \log t(n))$ | $O(\log t(n))$ (amortized) |
| Qiu et al. [2024] | $O(\log t(n))$ | $O(1)$ | $O(t(n) \log t(n))$ | $O(\log t(n))$ (amortized) |
| **This work** | $O(1)$ | $O(1)$ | $s(n)$ | $s(n)$ |

* Any multi-tape TM running in time $t(n)$ can be simulated by a Boolean circuit of size $O(t(n) \log t(n))$ [Arora and Barak, 2009].

AGI [Lecun, 2024, Goldblum et al., 2024], particularly for lifelong learning agents that interact with an open environment and accumulate an ever-growing history (e.g., the AIXI agent [Hutter, 2005]), where the input length grows unboundedly over time.

This motivates the following critical question:

*Problem 1: Is it necessary to continue scaling up the bit size of transformers to handle longer inputs?*

## 1.1 Our contribution

We answer Problem 1 in the negative. We prove that *constant bit-size* transformers[2] are Turing complete. Specifically, given any TM, there exists a transformer with fixed numerical precision and a fixed number of parameters that can simulate the TM on inputs of arbitrary length, provided that the transformer's context window is sufficiently long. In particular, by applying it to the universal Turing machine, which takes as input a description TM of a TM and an input string $x$ and outputs $TM(x)$, we can conclude that a single, constant bit-size transformer can compute any computable function, as long as the description of a relevant TM is loaded in the prompt.

**Remark 1.** *In fact, it is not even necessary to load the task-specific TM description into the prompt or to pre-inject it during pre-training. Instead, the transformer can be instantiated to simulate a TM that performs the meta-task of designing an algorithm for the given reasoning problem. For example, one can design the transformer to simulate Levin's universal search algorithm [Levin, 1973, 1984, Schmidhuber, 2004], which solves arbitrary search problems, such as theorem proving and planning, as quickly as the fastest possible algorithm in an asymptotic sense. This suggests that, in principle, a single constant bit-size transformer has sufficient expressive power to achieve a form of general reasoning ability.*

We emphasize that the context window length is the minimal resource needed to scale up to handle longer inputs: The context window must be at least as long as the input length $n$, otherwise the transformer cannot even access the entire input. Moreover, we provide an exact characterization of the expressive power of constant bit-size transformers as a function of their context window length. Specifically, let $\mathsf{WINDOW}[s(n)]$ denote the complexity class consisting of decision problems solvable by a constant bit-size transformer with a context window of length $O(s(n))$, $\mathsf{SPACE}[s(n)]$ the class of decision problems solvable by a TM using $O(s(n))$ space, and $\mathsf{PSPACE} := \bigcup_{k \in \mathbb{N}} \mathsf{SPACE}[n^k]$ the class of decision problems solvable in polynomial space. We now state the main theorem of this paper:

**Theorem 1.** *For any non-decreasing function $s(n) \geq n$, we have $\mathsf{WINDOW}[s(n)] = \mathsf{SPACE}[s(n)]$. In particular, $\mathsf{WINDOW}[\mathrm{poly}(n)] = \mathsf{PSPACE}$.*

Previous proofs establishing the Turing completeness of transformers suggested that the context window length must scale with the time complexity for solving the task. In contrast, Theorem 1 implies that it is sufficient for the context window length to scale only with the space complexity instead. This distinction is crucial, since many problems exhibit a significant gap between space

---

[2]Transformers are allowed to generate arbitrarily long CoTs before producing the final output.

complexity and time complexity. For instance, solving Boolean satisfication problems, Sokoban puzzles [Hearn and Demaine, 2005], model checking tasks [Sistla and Clarke, 1985], or other PSPACE-complete problems, require only polynomial space by tracking the current configuration, whereas the time complexity is widely conjectured to be superpolynomial [Arora and Barak, 2009] or even $2^{\mathsf{poly}(n)}$ [Impagliazzo and Paturi, 2001]. Recently, Williams [2025] proved that every multi-tape Turing machine running in time $t(n)$ can be simulated in space only $O(\sqrt{t(n)\log t(n)})$. As a consequence, there are problems solvable in $O(s(n))$ space that require $\tilde{\Omega}(s(n)^2)$ time on multitape Turing machines.

In addition, the simulation of a TM by a transformer in Theorem 1 is uniform and easily generalizable with respect to the context window length. Specifically, to simulate a given TM on increasingly long inputs, the transformer just needs to extend its context window by adjusting the relative positional encoding according to an explicit formula, without changing all other model parameters.

**Remark 2.** *Aiming for stronger deductive reasoning ability, a line of engineering efforts tend to lengthen context window instead of growing parameter count. Our result formally justifies this strategy: extending the window alone is sufficient. Moreover, a window that grows only polynomially with the input already suffices to solve every PSPACE problem, including game-tree search and mathematical proofs.*

**Remark 3.** *There is a ongoing debta on whether transformers can approach AGI. A common negative view (e.g. [Lecun, 2024, Goldblum et al., 2024]) argues that transformers, limited by finite size and window, cannot approximate unbounded computation. Our result shows that, at least for deductive reasoning tasks (whose solutions depend only on explicit information), scaling window length alone is theoretically sufficient, and continually scaling up size is not a principled requirement.*

Our proof strategy is to simulate Post machines [Post, 1936] with transformers. A Post machine can be modeled as an automaton equipped with a queue. By storing the TM's tape within the queue in a cyclic manner, Post machines can faithfully simulate Turing machines, and thus are Turing complete [Post, 1936, Davis et al., 1994]. By viewing the transformer's context window as a queue-like structure, one can observe that the behavior of Post machines naturally aligns with that of transformers. Leveraging this behavioral similarity, we can then design a transformer that faithfully simulates the step-by-step execution of a Post machine. In general, this alignment offers a new interpretation of the attention mechanism: not only as a statistical aggregator, but also as a discrete computational operation over queue-like structures.

## 1.2   Other related work

Recent theoretical studies have examined the computational capabilities of transformers under various architectural constraints. Feng et al. [2023], Merrill and Sabharwal [2023, 2024], Li et al. [2024] proved that CoT steps can improve the reasoning capabilities of transformers: Transformers without CoT can only solve problems that can be solved by shallow circuits, whereas allowing polynomially long CoT steps enables the same model to solve any problem in P. In addition, Merrill and Sabharwal [2025] showed that a transformer with depth growing only logarithmically in the input length can recognize regular languages and solve graph connectivity problems without CoT, whereas a constant-depth transformer requires CoT steps of length $\Omega(n)$. We remark that these papers assume either a $O(\log n)$-bit precision or a $O(\log n)$-embedding size, and require the context window to be long enough to cover the entire context. The survey by Strobl et al. [2024] provides a comprehensive overview on formal-language expressivity of transformers, clarifying how design choices affect expressivity.

Aiming to reduce the required context window length and the memory, Yang et al. [2025] proposed the PENCIL framework, which incorporates a reduction mechanism to recursively "erase" intermediate CoT steps. This allows the context window length to scale with the space complexity of the computation, rather than the time complexity. We remark that their results still require increasing the bit-size of the transformer as the input length grows.

Schuurmans et al. [2024] considered a generalization of autoregressive decoding where, given a long input, emitted tokens are appended to the end of the sequence as the context window advances, and proved that the resulting system is Turing complete. This work also exploits the similarity between the behaviors of Post machines and Transformers. In fact, they analyze an even restrictive model, the so-called Lag system, where the next token depends solely on the leftmost **two** tokens in the queue

and **not** on the state $q$ of PM. The main technical challenge in their construction is thus to compensate for the absence of access to the state $q$. As a result, their simulation requires $\Theta(s(n)^3)$ CoT tokens per TM step, whereas ours are only $O(s(n))$.

## 2 Notations and preliminaries

Let $\mathbb{R}$ be the set of real numbers and $\mathbb{N}$ be the set of natural numbers. For $n \in \mathbb{N}$, let $[n]$ denote $\{1, 2, \cdots, n\}$. We use bold lowercase letters to represent vectors or sequences (e.g., $\boldsymbol{x}$), and bold uppercase letters to represent matrices (e.g., $\boldsymbol{A}$). Given a vector $\boldsymbol{x}$ and indices $j \leq i$, we use $\boldsymbol{x}_{j:i}$ to denote the sub-vector $(x_j, x_{j+1}, \ldots, x_i)$. Let $\boldsymbol{0}_n$ denote the $n$-dimensional all-zeros vector. Given a vocabulary $\mathcal{V}$, let $\mathcal{V}^n$ denote the set of length-$n$ strings over $\mathcal{V}$, and let $\mathcal{V}^* := \bigcup_{n=1}^{+\infty} \mathcal{V}^n$ denote the set of all finite strings over $\mathcal{V}$.

### 2.1 Turing machines

A (single-tape) Turing machine (TM) is defined as a tuple $\langle \Sigma, Q, \delta \rangle$ where

- $\Sigma$ is the tape alphabet, including a designated "blank" symbol $\bot$, a designated "start" symbol $\triangleright$, and numbers 0 and 1.
- $Q$ is a finite set consisting of possible states. We assume $Q$ contains a designated start state $q_{start}$ and a designated halting state $q_{halt}$.
- $\delta : Q \times \Sigma \to Q \times \Sigma \times \{\text{Left}, \text{Right}\}$ is a transition function describing the rules used in performing each step of the TM.

The machine has a single tape that is infinite to the right and bounded on the left. The tape is equipped with a tape head. Initially, the tape (from the leftmost cell rightward) contains the start symbol $\triangleright$, a finite $\{0, 1\}$-string $x$ (the input), and the blank symbols $\bot$ on the rest of its cells. The head starts at the leftmost tape cell, and the machine starts in the state $q_{start}$. The computation then repeats the following step until the machine enters $q_{halt}$: if the machine is in some state $q \in Q$ and is reading symbol $\sigma$ from the tape, and if $\delta(q, \sigma) = (q', \sigma', z)$ for some $z \in \{\text{Left}, \text{Right}\}$, then the machine replaces $\sigma$ with $\sigma'$ in the current cell, changes state from $q$ to $q'$, and moves the tape head one cell in direction $z$.

**Space complexity** Let $s : \mathbb{N} \to \mathbb{N}$ be a non-decreasing function. The complexity class $\mathsf{SPACE}[s(n)]$ consists of all decision problems $\{0, 1\}^* \to \{0, 1\}$ that can be decided by a single-tape Turing machine using at most $O(s(n))$ tape cells on inputs of length $n$.[3] The class $\mathsf{PSPACE} := \bigcup_{k \in \mathbb{N}} \mathsf{SPACE}[n^k]$ is defined as the class of decision problems that can be solved in polynomial space.

### 2.2 Post machines

A Post machine (PM) [Post, 1936] is defined as a tuple $\langle \Sigma, Q, \delta \rangle$ where

- $\Sigma$ is the tape alphabet, including a designated "blank" symbol $\bot$, a designated "start" symbol $\triangleright$, and numbers 0 and 1.
- $Q$ is a finite set consisting of possible states. We assume $Q$ contains a designated start state $q_{start}$ and a designated halting state $q_{halt}$.
- $\delta : Q \times \Gamma \to Q \times \Gamma \times \{\text{Stay}, \text{Right}\}$ is the transition function, specifying how the machine updates the tape and state.

The machine has a single tape that is infinite to the right and bounded on the left. The tape is equipped with two tape heads: the *front head* and the *rear head*. Initially, the tape (from leftmost cell rightward) contains the start symbol $\triangleright$, a finite $\{0, 1\}$-string $x$ (the input), and the blank symbols $\bot$ on the rest of its cells. The front head starts at the leftmost tape cell, the rear head starts at the right end of the input, and the machine starts in state $q_{start}$. The computation repeats the following step until the machine

---

[3]We implicitly assume that $s(n) \geq n$, since $n$ tape cells are required to load an input of length $n$.

enters $q_{halt}$: if the machine is in state $q \in Q$ with the front head reading symbol $\sigma$ from the tape, and if $\delta(q, \sigma) = (q', \sigma', z)$ for some $z \in \{\text{Stay}, \text{Right}\}$, then the machine (i) changes the state from $q$ to $q'$, (ii) moves the front head either one cell to the right if $z = \text{Right}$, and (iii) moves the rear head one cell to the right, and then write $\sigma'$.

One can see that a PM can also be modeled as an automaton equipped with a queue of unbounded size: the queue is initialized as the input $x$, the front head points to the first element of the queue, and the rear head points to the last one. In each step, the PM reads and deletes the first element of the queue and may append a string to the tail.

**Turing completeness of PM**  The model of PM is Turing complete, i.e., equivalent in computational power to TMs. Specifically,

**Theorem 2.** *Given any single-tape TM running in $t(n)$ time and $s(n)$ space, there is an equivalent PM running in $O(t(n) \times s(n))$ time and using a $s(n)$-size queue.*

This theorem has been implicitly proved in several places, e.g., [Umans, 2025] and Section 9.1 of [Pettorossi, 2014]. For completeness, we present a proof in Appendix B. The intuition is as follows. The PM's queue stores the non-blank part of the TM's tape in a cyclic manner: If the TM head changes a symbol $\sigma$ to $\sigma'$ and moves right, the queue shifts by deleting $\sigma$ from the front and appending $\sigma'$ to the end. If the TM head moves left, the queue cyclically rotates by moving the last symbol to the front, where the appended symbol may need to depend on the leftmost **two** queue symbols. To avoid explicitly pre-reading the next queue cell, we apply a one-step *delayed append* trick: initially, pop the leftmost queue symbol and store it in the finite-state controller, without appending; thereafter, at each step, (i) pop the next leftmost symbol, and (ii) append a symbol determined by the two most recently popped symbols.

## 2.3  Transformers

For simplicity of notation, we present the definition of a single-head transformer, since only single-head attention is used in our proof. The definition of general multi-head transformers is deferred to Appendix C.

Let $\mathcal{V}$ be a finite vocabulary. A decoder-only transformer is a parameterized neural network $\mathsf{TF}_\theta$ mapping from $\mathcal{V}^*$ to $\mathcal{V}$. Specifically, a transformer is a composition $\mathsf{TF} := \mathsf{out} \circ \mathsf{dec}_{L-1} \circ \cdots \circ \mathsf{dec}_0 \circ \mathsf{pos} \circ \mathsf{emb}$ of four kinds of layers. Given a sequence $\boldsymbol{v} = (v_1, \cdots, v_i) \in \mathcal{V}^i$, it computes the next token $v_{i+1}$ as follows:

**1. Token embedding layer** (emb): For $j \leq i$, map each $v_j \in \mathcal{V}$ to a vector $\mathsf{emb}(v_j) \in \mathbb{R}^d$. Here, $d$ is called the *embedding size*.

**2. Positional encoding layer** (pos): For $j \leq i$, add a positional encoding $\mathsf{pos}(i - j) \in \mathbb{R}^d$ to the token embedding $\mathsf{emb}(v_j)$, resulting in the initial input representation $\boldsymbol{h}_j^0 := \mathsf{emb}(v_j) + \mathsf{pos}(i - j)$. Here, we assume a relative positional encoding scheme, where the encoding depends only on the distance between $v_j$ and $v_i$.

**3. Decoder Layer** ($\mathsf{dec}_\ell$ **for** $\ell = 0, \cdots, L - 1$): Each decoder layer $\mathsf{dec}_\ell$ consists of two sublayers: an self-attention layer, followed by a fully-connected feed-forward network. Residual connections are applied around each sub-layer [4]. Here, following common practice Pérez et al. [2021], Merrill and Sabharwal [2024, 2023], Qiu et al. [2024], Yang et al. [2025], we use hardmax as a realistic abstraction of softmax. In fact, hardmax is the instance-wise limit of zero temperature limit of softmax [Yang et al., 2025]. Consider a vector $\boldsymbol{s} \in \mathbb{R}^n$. If the maximum value in $\boldsymbol{s}$ appears at $t$ positions, then the hardmax function is defined coordinate-wise as:

$$\mathsf{hardmax}(\boldsymbol{s})_j := \begin{cases} \frac{1}{t}, & \text{if } s_j = \max_k s_k, \\ 0, & \text{otherwise.} \end{cases}$$

*3.1. Single-head self-attention layer*: Compute the attention score

$$\boldsymbol{s}_i^\ell = \mathsf{hardmax}\left( \langle \boldsymbol{h}_1^\ell \cdot \boldsymbol{Q}^\ell, \boldsymbol{h}_i^\ell \cdot \boldsymbol{K}^\ell \rangle, \cdots, \langle \boldsymbol{h}_i^\ell \cdot \boldsymbol{Q}^\ell, \boldsymbol{h}_i^\ell \cdot \boldsymbol{K}^\ell \rangle \right),$$

---

[4]For simplicity, we ignore layer normalizations from the standard transformer architecture [Radford et al., 2019] in our analysis. With a little more technical treatment as in [Li et al., 2024], our results can be extended to transformers with layer normalizations.

and then

$$\boldsymbol{a}_i^\ell = \sum_{j=1}^i s_{i,j}^\ell \cdot v^\ell(\boldsymbol{h}_i^\ell), \text{ where } v^\ell(h) := \boldsymbol{h} \cdot \boldsymbol{V}^\ell \text{ and } s_{i,j}^\ell \text{ is the } j\text{-th entry of } \boldsymbol{s}_i^\ell.$$

Here, $\boldsymbol{Q}^\ell, \boldsymbol{K}^\ell, \boldsymbol{V}^\ell \in \mathbb{R}^{d \times d}$ are parametrized matrices. Then, this sublayer returns

$$\boldsymbol{h}_i^{\ell+0.5} := \boldsymbol{W}^\ell \cdot \boldsymbol{a}_i^\ell + \boldsymbol{b}^\ell + \boldsymbol{h}_i^\ell,$$

where $\boldsymbol{W}^\ell \in \mathbb{R}^{d \times d}$ and $\boldsymbol{b}^\ell \in \mathbb{R}^d$ are parametrized.

*3.2. Feed-forward network*: Apply a multi-layer fully-connected ReLU neural network FF to $\boldsymbol{h}_i^{\ell+0.5}$, and returns
$$\boldsymbol{h}_i^{\ell+1} = \mathsf{FF}(\boldsymbol{h}_i^{\ell+0.5}) + \boldsymbol{h}_i^{\ell+0.5}.$$

**4. Output layer** (out): The final output representations $\boldsymbol{h}_i^L$ are projected onto the vocabulary space using a linear transformation followed by a $\arg\max$ function:

$$v_{i+1} := \mathsf{out}(\boldsymbol{h}_i^L) = \arg\max(W^{\mathsf{out}} \cdot \boldsymbol{h}_i^L + \boldsymbol{b}^{\mathsf{out}}), \text{ where } W^{\mathsf{out}} \in \mathbb{R}^{|\mathcal{V}| \times d} \text{ and } \boldsymbol{b}^{\mathsf{out}} \in \mathbb{R}^{|\mathcal{V}|}.$$

We say that a transformer operates with a *context window* of length $s$ if the generation of $v_{i+1}$ depends only on the last $s$ tokens $v_{i-s+1}, \cdots, v_i$. A transformer is said to be of $p$-bit precision if each parameter is represented using $p$ bits. We define the *bit-size* of a transformer $\mathsf{TF}_\theta$ as $p \times |\theta|$, meaning that the entire model can be represented by using $p \times |\theta|$ bits. In this paper, when we refer to transformers, we mean transformers that are allowed to generate arbitrarily long immediate CoT steps, unless stated otherwise.

**Definition 1.** *We define* $\mathsf{WINDOW}[s(n)]$ *as complexity class consisting of the problems* $\{0,1\}^* \to \{0,1\}$ *that can be solved by a constant bit-size transformer with context window of length* $O(s(n))$ *on inputs of length* $n$.

## 3 Proof of main results

In this section, we prove Theorem 1, which characterizes the expressive power of constant bit-size transformers with context window of length $s(n)$ in terms of the complexity class $\mathsf{SPACE}[s(n)]$. In particular, it implies that constant bit-size transformers are Turing complete.

First, we prove that any constant bit-size transformer with a context window of length $s(n)$ can be simulated by a Turing machine using $O(s(n))$ space.

**Theorem 3.** *For any non-decreasing function* $s(n) \geq n$, $\mathsf{WINDOW}[s(n)] \subseteq \mathsf{SPACE}[s(n)]$.

*Proof.* Let $L : \{0,1\}^* \to \{0,1\}$ be a decision problem that can be solved by a constant bit-size transformer using a context window of length $s(n)$. We sketch a Turing machine using $O(s(n))$ space that solves $L$ by simulating the transformer. The Turing machine maintains a buffer of size $s(n)$ storing the latest $s(n)$ tokens in the transformer's context. To compute the next token generated by the transformer, the Turing machine simulates the computation procedure described in Section 2.3, using an additional $O(s(n))$ working space. After generating the next token, the Turing machine updates the buffer by appending the new token and deleting the oldest token that slides out of the window, and then cleans the working space. By iteratively repeating this procedure, one can see that the Turing machine can faithfully simulate the transformer's behavior and solve the problem $L$. $\square$

In the following, we prove the reverse direction, which is the main technical part.

**Theorem 4.** *Let* $\mathsf{TM}$ *be a single-tape Turing machine that, on input* $x \in \{0,1\}^n$, *uses at most* $s(n)$ *space and runs for at most* $t(n)$ *steps. There exists a constant bit-size transformer with a context window of length* $O(s(n))$ *that, on input* $x$, *takes* $O(t(n) \cdot s(n))$ *CoT steps and then outputs* $\mathsf{TM}(x)$.

*As a consequence, we have* $\mathsf{WINDOW}[(s(n))] \supseteq \mathsf{SPACE}[s(n)]$ *for* $s(n) \geq n$.

*Proof.* Let $\mathsf{TM}$ be a single-tape Turing machine that runs in $t(n)$ steps and uses at most $s(n)$ tape cells. By Theorem 2, the $\mathsf{TM}$ can be simulated by a Post machine that runs in $O(t(n) \cdot s(n))$ time and

uses a queue of size $s(n)$. We slightly adapt the Post machine to ensure that the queue size remains exactly $s(n)$ throughout the computation, or equivalently, that the distance between the front and rear heads on the tape remains fixed at $s(n)$. Specifically, we pad the input $x$ with $s(n) - n$ copies of a special symbol $\#$ to the right initially, and in each computation step, both the front and rear heads always move one cell to the right. One can easily see that such an adapted PM $= (\Sigma, Q, \delta)$ simulates the Turing machine as well. Furthermore, without loss of generality, we assume $\Sigma = \{0, 1, \#\}$, $Q = \{0, 1\}^c$ for some $c \in \mathbb{N}$, and that $q_{start} \notin \delta(\sigma, q)$ for any $(\sigma, q)$, meaning that once the computation starts, the Post machine leaves $q_{start}$ immediately and never comes back.

We now construct a constant bit-size transformer TF with a context window of length $s(n)$ to faithfully simulate PM. The intuition is as follows:

- The $s(n)$-size queue is simulated by the $s(n)$-long context window, where the first element in the queue (or the tape cell pointed by the front head) corresponds to the oldest token in the window, and the last element in the queue (or the tape cell pointed by the rear head) corresponds to the newest token. Besides, we will let the vocabulary of the transformer be $\mathcal{V} = \Sigma \times Q$, so that a token also tracks the PM state information.

- As mentioned above, in each computation step of PM, the first element in the queue is removed, and a new element is added, so that the queue size remains the same. Correspondingly, in each CoT step of the transformer, the oldest token in the window slides out, and a new token is appended.

- The transition function of PM takes the last element in the queue and the current state as input, and outputs the next state and the next element. Correspondingly, by carefully choosing the parameters, the self-attention layer retrieves the last element from the oldest token in the window, and the current state from the current token; subsequently, a feed-forward network is used to implement the transition function $\delta$.

We formally define the transformer TF as follows: let $\mathcal{V} = \Sigma \times Q = \{0, 1, \#\} \times \{0, 1\}^c$.

1. Token embedding layer    Map each token $v = (\sigma, q)$ to a vector $\mathsf{emb}(\sigma, q) \in \mathbb{R}^{c+8}$ as follows:

$$\mathsf{emb}(\sigma, q) = \begin{cases} (1, 1, 0, 0, q, 0, 0, 0, 0), & \text{if } \sigma = 0; \\ (1, 0, 1, 0, q, 0, 0, 0, 0), & \text{if } \sigma = 1; \\ (1, 0, 0, 1, q, 0, 0, 0, 0), & \text{if } \sigma = \#. \end{cases} \tag{1}$$

2. Positional encoding layer    Add a relative positional encoding $\mathsf{pos}(i - j) \in \mathbb{R}^{c+8}$ to the token embedding $\mathsf{emb}(v_j)$ where

$$\mathsf{pos}(i - j) = \begin{cases} \mathbf{0}_{c+8}, & \text{if } 0 < i - j < s(n) - 1; \\ (\mathbf{0}_{c+7}, 1), & \text{if } i - j = 0, \\ (\mathbf{0}_{c+7}, 2), & \text{if } i - j = s(n) - 1, \end{cases} \tag{2}$$

Then return $\boldsymbol{h}_j^0 = \mathsf{emb}(v_j) + \mathsf{pos}(i - j)$.

3. Decoder layer    The transformer has only one decoder layer, and the decoder layer has only one attention head. In the self-attention layer, we set the matrix $\boldsymbol{K}, \boldsymbol{Q}, \boldsymbol{V}, \boldsymbol{W} \in \mathbb{R}^{d \times d}$ and $\boldsymbol{b} \in \mathbb{R}^d$ where $d = c + 8$ as follows:

- Matrix $\boldsymbol{K}$: it has only one non-zero entry $K_{c+8,1} = 1$;

- Matrix $\boldsymbol{Q}$: it has only one non-zero entry $Q_{c+8,c+8} = 1$;

- Matrix $\boldsymbol{V}$: it has four non-zero entries: $V_{c+5,2} = V_{c+6,3} = V_{c+7,4} = V_{c+8,c+8} = 1$.

Besides, we set the matrix $\boldsymbol{W} \in \mathbb{R}^{d \times d}$ to be the identity matrix, and $\boldsymbol{b} = \mathbf{0}_{c+8}$.

The feed-forward network FF is designed to simulate transition function $\delta$. Specifically, we let

$$\mathsf{FF}(\boldsymbol{h}) + \boldsymbol{h} = \begin{cases} \boldsymbol{e}_{(\#,q_{start})}, & \text{if } h_{c+8} = 2; \\ \boldsymbol{e}_{\delta(0,q)}, & \text{if } \boldsymbol{h}_{4:c+8} = (q,1,0,0,3); \\ \boldsymbol{e}_{\delta(1,q)}, & \text{if } \boldsymbol{h}_{4:c+8} = (q,0,1,0,3); \\ \boldsymbol{e}_{\delta(\#,q)}, & \text{if } \boldsymbol{h}_{4:c+8} = (q,0,0,1,3); \end{cases} \tag{3}$$

Here, $\boldsymbol{e}_{(\sigma',q')} \in \mathbb{R}^{|\mathcal{V}|}$ is the unit vector where the entry indexed by $(\sigma',q')$ is 1 and any other entry is 0.

4. Output layer  We set $W^{\mathsf{out}}$ to be the identity matrix and $\boldsymbol{b}^{\mathsf{out}}$ the all-zeros vector. Then the output layer outputs $v_{i+1} := \arg\max(\boldsymbol{h}_i^1)$.

In the following, we show that the transformer TF faithfully simulates the Post machine PM. Let $(\sigma_1,q_1),(\sigma_2,q_2),\cdots$ denote the execution log of PM running on $x \in \{0,1\}^n$, where $\sigma_i$ is the $i$-th element added to the queue and $q_i$ is the state when adding $\sigma_i$. Note that $(\sigma_i,q_i) = (x_i,q_{start})$ for $i \leq n$, $(\sigma_i,q_i) = (\#,q_{start})$ for $n+1 \leq i < s(n)$, and $\delta(\sigma_{i-s(n)+1},q_i) = (\sigma_{i+1},q_{i+1})$ for $i \geq s(n)$. Let $v_1, v_2, \ldots$ denote the token sequence in the context of TF when it takes $v_1 = (x_1, q_{start}), \ldots, v_n = (x_n, q_{start})$ as input. We show by induction that for each $i \geq 1$, we have $v_i = (\sigma_i, q_i)$.

Base case: $i \leq n$  This case is trivial.

Inductive case: $n+1 \leq i < s(n)$  In the self-attention layer, observing that $\boldsymbol{K} \cdot \boldsymbol{h}_i^0 = (\boldsymbol{0}_{c+7}, 1) = 1$ and

$$\boldsymbol{Q} \cdot \boldsymbol{h}_j^0 = \mathsf{pos}(i-j) = \begin{cases} \boldsymbol{0}_{c+8}, & \text{if } 1 \leq j < i, \\ (\boldsymbol{0}_{c+7}, 1), & \text{if } j = i, \end{cases} \tag{4}$$

we have

$$\boldsymbol{s}_i = \mathsf{hardmax}\left(\langle \boldsymbol{h}_1^0 \cdot \boldsymbol{Q}, \boldsymbol{h}_i^0 \cdot \boldsymbol{K}\rangle, \cdots, \langle \boldsymbol{h}_i^0 \cdot \boldsymbol{Q}, \boldsymbol{h}_i^0 \cdot \boldsymbol{K}\rangle\right) = (0,\cdots,0,1) \in \mathbb{R}^i$$

and

$$\boldsymbol{a}_i = \sum_{j=1}^{i} s_{i,j} \cdot \left(\boldsymbol{V} \cdot \boldsymbol{h}_j^0\right) = \sum_{j=1}^{i} s_{i,j} \cdot \left(\boldsymbol{0}_{c+4}, \boldsymbol{h}_{j,2:4}^0, h_{j:c+8}^0\right) = (\boldsymbol{0}_{c+4}, \boldsymbol{h}_{i,2:4}^0, 1).$$

Then

$$\boldsymbol{h}_i^{0.5} = \boldsymbol{W} \cdot \boldsymbol{a}_i + \boldsymbol{b} + \boldsymbol{h}_i^0 = \boldsymbol{a}_i + \boldsymbol{h}_i^0 = (\boldsymbol{h}_{i,1:c+4}^0, \boldsymbol{h}_{i,2:4}^0, 2).$$

In the feed-forward network layer, $\boldsymbol{h}_i^{0.5}$ is mapped to $\mathsf{FF}(\boldsymbol{h}_i^{0.5}) + \boldsymbol{h}_i^{0.5}$, which is $\boldsymbol{e}_{(\#,q_{start})}$ according to Equation (3). Finally, the output layer outputs $(\#, q_{start})$ as desired.

Inductive case: $i \geq s(n)$  In the self-attention layer, observing that $\boldsymbol{K} \cdot \boldsymbol{h}_i^0 = (\boldsymbol{0}_{c+7}, 1) = 1$ and

$$\boldsymbol{Q} \cdot \boldsymbol{h}_j^0 = \mathsf{pos}(i-j) = \begin{cases} 0, & \text{if } i - s(n) + 1 < j < i; \\ 1, & \text{if } j = i, \\ 2, & \text{if } j = i - s(n) + 1, \end{cases} \tag{5}$$

we have

$$\boldsymbol{s}_i = \mathsf{hardmax}\left(\langle \boldsymbol{h}_{i-s(n)+1}^0 \cdot \boldsymbol{Q}, \boldsymbol{h}_i^0 \cdot \boldsymbol{K}\rangle, \cdots, \langle \boldsymbol{h}_i^0 \cdot \boldsymbol{Q}, \boldsymbol{h}_i^0 \cdot \boldsymbol{K}\rangle\right) = (1, 0\cdots, 0) \in \mathbb{R}^{s(n)}$$

and

$$\boldsymbol{a}_i = \sum_{j=i-s(n)+1}^{i} s_{i,j} \cdot \left(\boldsymbol{0}_{c+4}, \boldsymbol{h}_{j,2:4}^0, h_{j:c+8}^0\right) = (\boldsymbol{0}_{c+4}, \boldsymbol{h}_{i-s(n)+1,2:4}^0, 2).$$

Then

$$\boldsymbol{h}_i^{0.5} = \boldsymbol{a}_i + \boldsymbol{h}_i^0 = (\boldsymbol{h}_{i,1:c+4}^0, \boldsymbol{h}_{i-s(n)+1,2:4}^0, 3).$$

In the feed-forward network layer, noting that by the induction hypothesis, we have

$$\boldsymbol{h}_{i,5:c+4}^{0.5} = \boldsymbol{h}_{i,5:c+4}^0 = q_i, \text{ and } \boldsymbol{h}_{i,c+5:c+7}^{0.5} = \boldsymbol{h}_{i-s(n)+1,2:4}^0 = \sigma_{i-s(n)+1}.$$

The vector $\boldsymbol{h}_i^{0.5}$ is mapped to $\mathsf{FF}(\boldsymbol{h}_i^{0.5}) + \boldsymbol{h}_i^{0.5}$, which is $\boldsymbol{e}_{\delta(\sigma_{i-s(n)+1}, q_i)}$ according to Equation (3). Recall that $\delta(\sigma_{i-s(n)+1}, q_i) = (\sigma_{i+1}, q_{i+1})$, then one can see that the output layer outputs $(\sigma_{i+1}, q_{i+1})$ as desired.

Now, we have shown that $v_i = (\sigma_i, q_i)$ for any $i \geq 1$ and thus finished the proof. $\qquad\square$

**Remark 4.** *As we can see from the proof, to simulate the given TM on longer inputs, what we need to do is just change the relative positional encoding according to Equation (2), without changing all other model parameters.*

**Remark 5.** *We briefly compare our construction with existing Turing-completeness proofs and highlight why our construction achieves constant bit-size. In [Pérez et al., 2021] and related follow-ups, the query may attend to many earlier tokens; when multiple positions tie for the maximum score, the hardmax outputs fractions such as $1/t$, which requires the precision to grow with t. The proof of Li et al. [2024] encodes the simulated circuit directly into the absolute positional encoding, rather than into the parameters. Consequently, as the circuit expands, the positional encoding must also grow in dimension, so the embedding size is no longer constant.*

*In contrast, our construction leverages the automaton-like behavior of Post machines, which is simpler and more regular than the head movements of Turing machines or the wiring patterns of circuits. As a result, at each CoT step the query attends to exactly one token, specifically the token located $s(n)$ positions earlier. This fixed-offset attention can be implemented purely through a relative positional encoding, without any additional arithmetic operations.*

*In particular, the input to* hardmax *is always one-hot, so is the resulted attention distribution. As an alternative of* hardmax*, one could replace* hardmax *with the right-most hardmax function [Yang et al., 2024].*

Combining Theorems 3 and 4, we immediately conclude our main theorem.

**Theorem 1.** *For any non-decreasing function $s(n) \geq n$, we have* $\mathsf{WINDOW}[s(n)] = \mathsf{SPACE}[s(n)]$. *In particular,* $\mathsf{WINDOW}[\mathrm{poly}(n)] = \mathsf{PSPACE}$.

## 4 Conclusions and Discussions

In this work, we have shown that a constant bit-size transformer, with a sufficiently long context window, can simulate any Turing machine on arbitrarily long inputs. We further proved that the complexity class $\mathsf{SPACE}[s(n)]$ precisely characterizes the expressive power of constant bit-size transformers with a context window of length $s(n)$, suggesting that the context window length needs to scale only with the space complexity rather than the time complexity as suggested by previous results. Our proof leverages the behavioral similarity between transformers and Post machine, which offers new insights into the mechanism underlying the reasoning abilities of transformers. We list some future directions:

- *Simulation efficiency*: Our construction requires $O(t(n) \cdot s(n))$ CoT steps, which is potentially prohibitive for practical applications. It would be interesting and important to investigate whether this slowdown can be avoided without compromising optimality in other aspects.

- *Positional encodings*: Our construction employs an nonstandard relative positional encoding. It is open whether it can be replaced with fixed absolute positional encodings or other standard relative positional encodings. Moreover, our positional encoding explicitly depends on the assumed space upper bound, and it is unclear how this bound could be inferred automatically.

- *Learnability and empirical validation*: Our contribution is purely about expressiveness. Whether standard training procedures can learn such behavior is an important open problem. How well our construction behaves in practice also remains to be investigated.

## 5 Acknowledgments

The authors thank the anonymous NeurIPS reviewers for their valuable suggestions.

Qian Li's work was supported by Shenzhen-Hong Kong Science and Technology Innovation Cooperation Zone Project (No.HZQSWS-KCCYB-2024016). Yuyi Wang's work was supported by the Hunan Provincial Natural Science Foundation (Grant No.2024JJ5128).

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

## A  Modifications on Transformers in Turing Completeness Proofs

Existing proofs of Turing-completeness for Transformers typically departure from the vanilla architecture and make specific modifications. Pérez et al. [2021] consider a Transformer with an absolute positional encoding given by the triplet $(i, 1/i, 1/i^2)$ at position $i$. In their construction, the attention score is defined as the negative absolute value of the dot product, and the attention mechanism uses average-hard attention. Moreover, the feed-forward layers employ sigmoid activations in place of ReLUs. Merrill and Sabharwal [2024] dispense with positional encodings altogether and instead adopt *Strict Causal Masking*, where attention at position $i$ can access all tokens up to position $i - 1$ but not the current token $i$. Their construction also uses average-hard attention and introduces a *Projected Pre-Norm*: an arbitrary linear projection is allowed before layer normalization. In the proof of Li et al. [2024], the simulated circuit is directly encoded into the absolute positional encoding, rather than into the parameters. Qiu et al. [2024] consider Transformers with nonstandard absolute positional encodings. In their construction, the query, key, and value maps in the attention sublayer are implemented as ReLU networks rather than linear transformations. This work employ nonstandard *relative* positional encodings.

## B  Proof of Theorem 2

Let $\mathsf{TM} = (\Sigma, Q, \delta)$ be a single-tape Turing machine that runs in time $t(n)$ and space $s(n)$. We construct an equivalent Post machine $\mathsf{PM} = (\Sigma', Q', \delta')$ that simulates $\mathsf{TM}$ within $O(t(n) \cdot s(n))$ time and uses a queue of size at most $s(n)$.

**Queue alphabet**  We define the queue alphabet $\Sigma' := \{\sigma, \hat{\sigma}, \tilde{\sigma} \mid \sigma \in \Sigma\}$. Here, $\hat{\ }$ and $\tilde{\ }$ are temporary markers used for simulating a left move of $\mathsf{TM}$'s head. Specifically, $\hat{\ }$ temporarily marks the current head cell, while $\tilde{\ }$ temporarily marks its left neighbor.

**Delayed append trick**  PM maintains a *logical queue* $L$ which represents the non-blank segment of $\mathsf{TM}$'s tape in a cyclic manner. Formally, we realize $L$ as $L = R \circ Q_{\text{phys}}$ where

- $R$ is a register in the finite control storing the leftmost symbol of $L$, and
- $Q_{\text{phys}}$ is the actual queue content. Thus $|Q_{\text{phys}}| = |L| - 1$.

With this representation, the appended symbol is no longer restricted to depend only on the leftmost symbol of $L$; it can now be determined by the two leftmost symbols.

**Initialization**  Initially, $\mathsf{TM}$'s tape contains the start symbol $\triangleright$, followed by the input $x$, and then blanks. Accordingly, the logical queue $L$ is initialized to $(\triangleright, x)$: the first symbol $\triangleright$ is placed in the register $R$, and the remaining symbols $x$ are stored into $Q_{\text{phys}}$. During the simulation, we will keep the following invariant:

- Right before each simulated step, the leftmost cell of $L$ correponds to $\mathsf{TM}$'s head cell; its symbol is either unmarked or carries the $\tilde{\ }$ mark, while all other cells of $L$ are unmarked.

**Simulation of one step**  Suppose $\mathsf{TM}$ is in state $q$ and its head reads $\sigma$ from cell $i$. Let $\delta(q, \sigma) = (q', \sigma', z)$ with $z \in \{\text{Left}, \text{Right}\}$. This means that $\mathsf{TM}$ overwrites cell $i$ with $\sigma'$, change state from $q$ to $q'$, and moves its head one cell in direction $z$.

- *Move Right*. PM deletes the leftmost logical symbol and appends $\sigma'$ to $L$, thereby overwriting tape cell $i$. If the resulting leftmost logical symbol is $\triangleright$, indicating that $\mathsf{TM}$ has extended its non-blank segment by one blank cell to the right, then PM additionally appends a blank symbol $\perp$.
- *Move Left*. PM deletes the leftmost logical symbol and appends $\hat{\sigma}'$ to $L$, thereby overwriting tape cell $i$. It then cyclically rotates $L$ until the second leftmost logical symbol carries the $\hat{\ }$ mark. At this point, the first and second of $L$ correspond to tape cells $i - 1$ and $i$, respectively. PM then adds a mark $\tilde{\ }$ to the first symbol and removes the $\hat{\ }$ mark from the second, after one additionally cyclic sweep of $L$; concretely,
  1. delete the leftmost element and append the same symbol with the mark $\tilde{\ }$; then

2. delete the leftmost element and append the same symbol with no mark; then

3. cyclically rotate $L$ until the leftmost logical symbol carries the $\tilde{\cdot}$ mark.

**Analysis**  By induction on the number of simulated steps, one can check that the invariant is maintained throughout the simulation, and hence PM faithfully simulate TM. For a right move, PM performs $O(1)$ queue operations. For a left move, it performs two full sweeps of $L$, costing $O(|L|) = O(s(n))$ operations. Therefore, the overall running time is $O(t(n) \cdot s(n))$, completing the proof.

## C  Multi-head Transformers

A multi-head Transformer employs the following *multi-head self-attention layer* in place of the single-head self-attention layer, while all other components remain the same as in the single-head transformer.

*Multi-head self-attention layer*: For each head $k = 1, 2, \cdots, H$, compute the attention score

$$\boldsymbol{s}_{k,i}^{\ell} = \mathsf{hardmax}\left(\langle \boldsymbol{h}_1^{\ell} \cdot \boldsymbol{Q}_k^{\ell}, \boldsymbol{h}_i^{\ell} \cdot \boldsymbol{K}_k^{\ell}\rangle, \cdots, \langle \boldsymbol{h}_i^{\ell} \cdot \boldsymbol{Q}_k^{\ell}, \boldsymbol{h}_i^{\ell} \cdot \boldsymbol{K}_k^{\ell}\rangle\right),$$

and then

$$\boldsymbol{a}_{i,k}^{\ell} = \sum_{j=1}^{i} s_{k,i,j}^{\ell} \cdot v_k^{\ell}(\boldsymbol{h}_i^{\ell}), \ \text{where } v_k^{\ell}(h) := \boldsymbol{h} \cdot \boldsymbol{V}_k^{\ell}$$

Here, $\boldsymbol{Q}_k^{\ell}, \boldsymbol{K}_k^{\ell}, \boldsymbol{V}_k^{\ell} \in \mathbb{R}^{d \times d/H}$ are parametrized matrices. By concatenating the $H$ heads, we obtain $\boldsymbol{a}_i^{\ell} = \left((\boldsymbol{a}_{i,1}^{\ell})^T, \cdots, (\boldsymbol{a}_{i,H}^{\ell})^T\right)^T \in \mathbb{R}^d$. Then, this sublayer returns

$$\boldsymbol{h}_i^{\ell+0.5} := \boldsymbol{W}^{\ell} \cdot \boldsymbol{a}_i^{\ell} + \boldsymbol{b}^{\ell} + \boldsymbol{h}_i^{\ell},$$

where $\boldsymbol{W}^{\ell} \in \mathbb{R}^{d \times d}$ and $\boldsymbol{b}^{\ell} \in \mathbb{R}^d$ are parametrized.

