# OpenReview forum: "Constant Bit-size Transformers Are Turing Complete"
_NeurIPS.cc/2025/Conference — NeurIPS 2025 poster_

### Official Review · Reviewer_XoFA · 2025-06-27

**Clarity:** 4
**Significance:** 3
**Originality:** 3
**Rating:** 5
**Confidence:** 4

**Summary:**

The paper is aimed at understanding the computational power of transformers with respect to the classical model of Turing Machines. Recent work has shown that transformers are Turing complete. This paper refines this body of work by showing that transformers of fixed size (in terms of size of embedding, number of layers, number of parameters) are sufficient. The equivalence proof does not require arbitrary precision arithmetic. It establishes precise relationship between the space bound of TM computation and the length of context window.

**Questions:**

The introduction highlights "fixed numerical precision" as a challenge w.r.t. earlier work. This is not discussed during the proof of the main result. It's likely obvious given the very discrete encoding, but could be explained (or more pertinently, why was this an issue in some of the earlier work, and what insight avoids it).

**Ethical Concerns:**

["NO or VERY MINOR ethics concerns only"]

**Final Justification:**

My original score was accept. The rebuttal and discussion was helpful to address any remaining questions. The final version is likely to be improved.

**Limitations:**

Yes

**Quality:**

4

**Strengths And Weaknesses:**

Strengths:
The main result (Theorem 1) is precise and concise. The authors make a compelling case that this is a tight result, in particular, allowing the context window length to grow with the input size is unavoidable.

The proof structure is very clear and intuitive. It uses a classical result that shows that TMs can be simulated by queue machines. They explain how a transformer can then simulate a queue machine: the context window stores the queue, adding elements corresponds to emitting next token. The main technical challenge is using the attention mechanism to retrieve the current state at the oldest symbol and simulate a transition.

Weakness:
The problem has been studied in a sequence of earlier papers. This is a nice and clean result, but doesn't seem surprising. Also not clear if this is relevant to practice. the claimed connection to understanding  "reasoning ability" of transformers is not justified.

---

> ### Author Rebuttal · Authors · 2025-07-29
>
> We sincerely thank the reviewer for the valuable feedback and questions. We will revised the related part according to the review.
>
>
>
> **Q1:** Also not clear if this is relevant to practice. The claimed connection to understanding "reasoning ability" of transformers is not justified.
>
> **A1:**  Our theorem establishes that: a single, fixed bit-size transformer can execute any algorithm whose working memory fits in the context window. It has several practical insights.
>
> - **Understanding the general reasoning capability.** Although our theorem is stated per task and may look like in the "one-model-one-task" paradigm at the first glance, it in fact leads to a universal-model perspective: a constant bit-size transformer can simulate a universal TM, i.e., a general-purpose programmable computer. Hence, a *single* specific model, once trained, can execute any program that is provided in the prompt or embedded into its parameters. This offers an interpretation on how current general-purpose LLMs handle diverse tasks.
>
>   As noted in Remark 1, the program need not to be task-specific; it could implement some meta-tasks, such as designing new algorithm.
>
> - **Engineering guidance.** Aiming for stronger deductive reasoning ability, a line of engineering efforts tend to lengthen context window instead of growing parameter count. Our result formally justifies this strategy: extending the window alone is sufficient. Moreover, a window that grows only polynomially with the input already suffices to solve every PSPACE problem, including game-tree search and mathematical proofs.
>
> - **Implications for the AGI debta.** There is a ongoing debta on whether transformers can approach AGI. A common negative view (e.g. [1,2]) argues that transformers, limited by finite size and window, cannot approximate unbounded computation.  Our result shows that, at least for deductive reasoning tasks (whose solutions depend only on explicit information), scaling window length alone is theoretically sufficient, and continually scaling up size is not a principled requirement.
>
>
>
> [1] Yann Lecun. Meta AI, open source, limits of LLMs, AGI & the future of AI. 2024
>
> [2] Goldblum et al. Position: The no free lunch theorem, Kolmogorov complexity, and the role of inductive biases in machine learning. 2024.
>
>
>
> **Q2:** The introduction highlights "fixed numerical precision" as a challenge w.r.t. earlier work. This is not discussed during the proof of the main result. It's likely obvious given the very discrete encoding, but could be explained (or more pertinently, why was this an issue in some of the earlier work, and what insight avoids it).
>
> **A2:**
> - **[Perez et al. 2021] and related follow-ups.**   The query may attend to many earlier tokens. When multiple positions tie for the maximal score, the hardmax outputs fractions such as $1/t$, which requires the precision to grow with $t$.
>
> - **[Li et al. 2024].** They encode the simulated circuit directly into the absolute PE (rather than into the parameters). As the circuit (hence the input length) expands, the PE must grow in dimension, so the embedding size is no longer constant.
>
> - **This work.** The Post machine’s automaton-like behavior is much simpler and more regular than the head-moving  behavior of TMs and the wiring of a circuit. This let our construction satisfy a nice property: at every CoT step the query attends to exactly one token, always the token $s(n)$ positions earlier. This fixed-offset attention can be realized solely by a relative positional encoding without any additional arithmetic operations. So the attention score is one-hot and entirely discrete; no averaging, division, or growing embedding is needed.

---

> > ### Comment · Reviewer_XoFA · 2025-08-01
> >
> > Thanks for the response. I now better understand the technical bits about fixed numerical precision.

---

### Official Review · Reviewer_Y1kW · 2025-06-30

**Clarity:** 4
**Significance:** 2
**Originality:** 2
**Rating:** 5
**Confidence:** 5

**Summary:**

Allowing CoT steps enables transformers with the computational power to perform complex reasoning tasks. A body of theoretical work has shed light on this phenomenon through proof that transformers can simulate Turing machines, under various architectural assumptions. This work investigates the setting where the precision and context window of the transformer must be bounded, and provides a novel proof of Turing-completeness using Post machines. They show that transformers with constant bit-size and context window s(n) are equivalent to SPACE[s(n)]. The computational behavior of Post machines aligns very well with CoT.

**Questions:**

Questions
- Is it possible to achieve the same effect of using the pos(i,j) positional encoding using some kind of an attention construction? While no longer constant bit-size, this would still be interesting in contrast with (https://arxiv.org/abs/2503.14337) by achieving space-efficiency without their reduction process.
- Other proofs also use non-standard positional encodings (like pos(i)=i^2 or 1/i), but in this case the positional encoding depending on s(n) seems even more nonstandard. Could the authors make these and other assumptions more clear?
- Is there more to say about the “natural alignment” between Post machines and transformers? Any future work or practical implications? There is a healthy amount of room left in the paper.

Suggestions
- It could be quite informative for the authors to create a table of various CoT Turing-completeness proofs to compare their result with others in the literature. For instance, the table could have columns for “precision” and “positional encoding” to clearly demonstrate the distinction with previous work.
- Technically, the definition of hardmax computing 1/t means that O(log n) bit precision is possible in the transformers. Your construction gets around this by enforcing that only one item in the context window has non-zero score. A note could be made about this, or you could use some form of argmax (like right-most hardmax https://arxiv.org/abs/2310.13897)
- It seems that [Liu et al. 2024] should be [Li et al. 2024]?
- There seems to be a typo on line 263 with the angle brackets

**Ethical Concerns:**

["NO or VERY MINOR ethics concerns only"]

**Final Justification:**

The authors addressed my concerns and provided an interesting discussion. My concern about computing S(n) was not completely assuaged, so I maintain my score of accept (5).

**Limitations:**

I think the authors could be more upfront about the positional encodings used and other assumptions they make, but otherwise things are pretty clear.

**Quality:**

3

**Strengths And Weaknesses:**

Strengths
- Very clearly written, constructions are shown in detail and I could follow them without much difficulty.
- The identification of Post machines as a natural analogue of CoT computation is very nice and I agree sheds more light on the behavior of transformers with CoT than the proofs with Turing machines.
- The work highlights the sometimes subtle differences between different algorithms that transformers may use in CoT


Weaknesses
- I’m not sure that the other proofs requiring an O(log n) precision is a very severe weakness that needs addressing – in practice the precision is often large compared to the sequence length (e.g. float16 can handle inputs on the order of 10^8 tokens without any problems (https://arxiv.org/pdf/2210.02671)).
- If I understand correctly, the construction crucially upon the relative positional encoding pos(i,j) which sweeps some things under the rug. While other works use non-constant bit precision in order to compute the token to attend to, the pos(i,j) simply provides location of the Post machine’s head, thus streamlining the process.
- For non-theory readers it would be informative to add a brief section on limitations or practical insights (given that you have 1.5 pages of space left over). For instance, what problems are solved in (say) an O(1) sized context window and how it compares to what problems transformers with CoT can solve in practice.

---

> ### Author Rebuttal · Authors · 2025-07-29
>
> We greatly appreciate the reviewer’s constructive comments and suggestions. We will polish the manuscript according to the review.
>
>
>
> **Q1:** I’m not sure that the other proofs requiring an O(log n) precision is a very severe weakness that needs addressing – in practice the precision is often large compared to the sequence length (e.g. float16 can handle inputs on the order of 10^8 tokens without any problems).
>
> **A1:** We agree that float16 or float32 is already sufficient for most current benchmarks. The issue could be relevant in lifelong continual learning setting: the agent may accumulate an ever-growing history, and the input length keeps increasing.
>
> In addition, proving that fixed precision suffices also provides a justification for the use of fixed-precision formats (and even aggressive quantization) in hardwares.
>
>
>
> **Q2:** If I understand correctly, the construction crucially upon the relative positional encoding pos(i,j) which sweeps some things under the rug. While other works use non-constant bit precision in order to compute the token to attend to, the pos(i,j) simply provides location of the Post machine’s head, thus streamlining the process.
>
> **A2:** The key ingredient is not the particular nonstandard relative PE itself but the following property of our construction:
>
> - **Fixed-offset attention:** At each step, the query attends to the only token exactly $s(n)$ positions earlier, which can be retrieved solely by a relative positional encoding without any other arithmetic operations.
>
> If we drop the constant bit-size requirement and switch to a fixed absolute PE, our second core result still holds: it remains sufficient for the context window length to scale with the space complexity. For example, one can use the absolute PE is $\mathrm{poi}(i)= i \mod p$ where $p>s(n)$. Because of the attention offset is always $s(n)$, the transformer can still retrieve the Post machine head using the absolute PE alone.
>
> In addition, the reviewer may see our response A4 to Reviewer aGq8 for an explanation of the key technical differences from previous proofs.
>
>
>
> **Q3**: For non-theory readers it would be informative to add a brief section on limitations or practical insights (given that you have 1.5 pages of space left over).
>
> Is there more to say about the “natural alignment” between Post machines and transformers? Any future work or practical implications? There is a healthy amount of room left in the paper.
>
> **A3:** We will add the following discussions on limitations and future directions:
>
> - CoT length: Our current simulation needs $O(t(n) s(n))$ CoT steps; reducing this to $O ⁣(t(n))$ remains an open problem.
> - Learnability and empirical validation: Our contribution is purely about expressiveness. Whether standard training procedures can learn such behavior is an important open problem. How well implementations behave in practice also remains to be investigated.
> - Positional encodings: Our construction employs an nonstandard relative positional encoding; extending the result to fixed absolute positional encodings or other standard relative positional encodings is open.
>
> We will add the following discussions on practical insights: Our theorem establishes that: a single, fixed bit-size transformer can execute any algorithm whose working memory fits in the context window. It has several practical insights.
>
> - **Understanding the general reasoning capability.** Although our theorem is stated per task and may look like in the "one-model-one-task" paradigm at the first glance, it in fact leads to a universal-model perspective: a constant bit-size transformer can simulate a universal TM, i.e., a general-purpose programmable computer. Hence, a *single* specific model, once trained, can execute any program that is provided in the prompt or embedded into its parameters. This offers an interpretation on how current general-purpose LLMs handle diverse tasks.
>
>   As noted in Remark 1, the program need not to be task-specific; it could implement some meta-tasks, such as designing new algorithm.
>
> - **Engineering guidance.** Aiming for stronger deductive reasoning ability, a line of engineering efforts tend to lengthen context window instead of growing parameter count. Our result formally justifies this strategy: extending the window alone is sufficient. Moreover, a window that grows only polynomially with the input already suffices to solve every PSPACE problem, including game-tree search and mathematical proofs.
>
> - **Implications for the AGI debta.** There is a ongoing debta on whether transformers can approach AGI. A common negative view (e.g. [1,2]) argues that transformers, limited by finite size and window, cannot approximate unbounded computation.  Our result shows that, at least for deductive reasoning tasks (whose solutions depend only on explicit information, such as Go or math), scaling window length alone is theoretically sufficient, and continually scaling up size is not a principled requirement.
>
> [1] Yann Lecun. Meta AI, open source, limits of LLMs, AGI & the future of AI. 2024
>
> [2] Goldblum et al. Position: The no free lunch theorem, Kolmogorov complexity, and the role of inductive biases in machine learning. 2024.
>
> Regarding the natural alignment between transformers and Post machines: the reviewer may see our response A4 to Reviewer aGq8 for an explanation on why simulating PMs leads to constant bit-size construction. An related open problem is whether the CoT length in our simulation be reduced from $O(t(n) s(n))$ to $O ⁣(t(n))$.
>
>
>
> **Q4:** For instance, what problems are solved in (say) an O(1) sized context window and how it compares to what problems transformers with CoT can solve in practice.
>
> **A4:** If the window length is $<n$, then the transformer cannot solve any problem that depends on the whole input, because the transformer cannot access the whole input.
>
> More specifically, note that the entire input string is presented before the transformer starts generating CoT tokens, and at every step the transformer can look back only $s(n)$ positions.  If $s(n)<n$, the first $n−s(n)$ input symbols are permanently outside the transformer's attention and can never influence any CoT token.
>
>
>
> **Q5:** Is it possible to achieve the same effect of using the pos(i,j) positional encoding using some kind of an attention construction? While no longer constant bit-size, this would still be interesting in contrast with "PENCIL: Long Thoughts with Short Memory" by achieving space-efficiency without their reduction process.
>
> **A5:** Yes. Because of the fixed-offset attention property (if we drop the constant bit-size constraint), the relative PE can be replaced by a fixed absolute PE (as mentioned in response A2), or a causal positional masking, which directly restrict token $i$ to only attend to the only token $i−s(n)$.
>
>
>
> Additionally, we will also revise other parts as suggested, including making the assumption of nonstandard relative PE more clear, fixing the typos, adding a note about hardmax, and adding a table comparing related results.

---

> > ### Comment · Reviewer_Y1kW · 2025-08-04
> > **Response to Authors**
> >
> > Thank you for your careful rebuttal and the interesting points you brought up! I will maintain my score because the paper is clearly-written and provides novel insight. While there are concerns about the practical relevance, I believe that does not diminish the contributions of this paper: providing a new perspective on how transformers may implement universal computation with bounded context size and precision. Whether the transformer actually uses this precise construction in practice is not particularly important to the insight. The connection to Post machines is novel, and situating this result with other CoT Turing completeness results will be helpful in future drafts. I have a remaining concern in my responses below:
> >
> > **A1 response**: The point is well taken.
> >
> > **A2 response**: My criticism essentially boils down to the opinion that the mechanism for the transformer to make its context window always size S(n) is too powerful. How would you implement enforcing an S(n) context window in a real transformer, without also endowing the transformer with the ability to simulate a post machine in order to compute S(n)?
> > It seems to me that other proofs of CoT Turing completeness could also become constant bit-size if we permit the use of positional encodings in this manner. For instance, [Perez et al. 2021] and related proofs required non-constant bit-size in order for the transformers to compute the head position. If we gave the transformer an absolute positional encoding that told it exactly where the head was at step n, then we could also make it constant bit-size.
> >
> > **A3 response**: This will be an interesting discussion!
> >
> > **A4 response**: Thank you! This makes sense.
> >
> > **A5 response**: I was essentially asking whether you could simulate the absolute PE using only attention and feed-forward layers. Computing S(n) does not seem computationally trivial to do, and I was wondering if this could be incorporated into the transformer.
> >
> > (Apologies, I had accidentally posted this under a different reviewer's thread and have now deleted it)

---

> > > ### Author Response · Authors · 2025-08-06
> > >
> > > **Q6:**  My criticism essentially boils down to the opinion that the mechanism for the transformer to make its context window always size S(n) is too powerful. How would you implement enforcing an S(n) context window in a real transformer, without also endowing the transformer with the ability to simulate a post machine in order to compute S(n)?
> > >
> > > I was essentially asking whether you could simulate the absolute PE using only attention and feed-forward layers. Computing S(n) does not seem computationally trivial to do, and I was wondering if this could be incorporated into the transformer.
> > >
> > > **A6:** Thank you for the clarification. We do not yet have a solution. We will list it as a further direction in the revision.
> > >
> > > **Q7:** For instance, [Perez et al. 2021] and related proofs required non-constant bit-size in order for the transformers to compute the head position. If we gave the transformer an absolute positional encoding that told it exactly where the head was at step n, then we could also make it constant bit-size.
> > >
> > > **A7:** If we understand correctly, you mean "relative PE" rather than "absolute PE", since such an absolute PE would need $\Omega(\log S(n))$ bit to encode the head location.
> > >
> > > Moreover, even if we allow non-standard relative PEs, we do not see how to achieve constant bit-size in [Perez et al. 2021] and related proofs, since the constructions do not satisfy the "fixed-offset attention" property.

---

> > > > ### Comment · Reviewer_Y1kW · 2025-08-07
> > > > **Response to Authors**
> > > >
> > > > Thank you for the response and clarification. I will maintain the accept score. I have no further questions and wish all the best to the authors!

---

### Official Review · Reviewer_AC2j · 2025-07-02

**Clarity:** 3
**Significance:** 3
**Originality:** 3
**Rating:** 5
**Confidence:** 3

**Summary:**

This paper is part of an active and growing line of theoretical research on the expressive power of transformer architectures. The central question is: how expressive are transformers as language deciders compared to traditional resource-bounded models like time- and space-bounded Turing machines (TMs)? This is a relatively new area with many open directions. A foundational result by Pérez et al. showed that certain transformer models are Turing complete when allowed to use Chain-of-Thought (CoT) reasoning, enabling them to simulate arbitrary computations. Since then, several works have explored what transformer parameter settings are necessary and sufficient to capture various classical models (e.g., space, time, Boolean circuit depth). A key parameter in this context is precision—the number of bits used to represent real numbers. Prior work established that logarithmic precision suffices for general computation.

This paper advances the field by showing that even constant-bit precision—a fixed number of bits—suffices to simulate Turing machines efficiently. Specifically, the authors define a transformer-based class ${\rm WINDOW}(s(n))$, where the ith token depends only on the preceding $s(n)$ tokens, and prove that ${\rm WINDOW}(s(n)) = {\rm SPACE}(s(n))$ for all $s(n) ≥ n$. Thus, constant-precision transformers with an attention window of size $O(s(n))$ recognize exactly the languages in ${\rm SPACE}(s(n))$.

**Questions:**

It would be interesting for the authors to explore and discuss the implications of their results when $s(n)$ is sublinear (e.g., $O(\log n))$. Does ${\rm WINDOW(O(log n))$ capture LOGSPACE? Are constant window transformers interesting?

**Ethical Concerns:**

["NO or VERY MINOR ethics concerns only"]

**Final Justification:**

The authors have promised to add more details and relation to prior work which was my earlier concern. I will update the score to 5.

**Limitations:**

There is no explicit Limitation section. But I am not concerned about that for this theoretical work.

**Paper Formatting Concerns:**

No Concerns,

**Quality:**

3

**Strengths And Weaknesses:**

Strengths: The paper makes a significant theoretical contribution by precisely characterizing the computational power of constant precision transformers in terms of classical complexity classes. It is indeed good to know that constant-bit precision suffices and this result is  relevant to both theory and practice.

Weaknesses: The literature review could be more thorough. In particular, the authors should look at the recent survey “What Formal Languages Can Transformers Express? A Survey” by Strobl et al. (https://arxiv.org/abs/2311.00208), which provides a comprehensive overview of formal language expressivity results for transformers. Comparing and contrasting your result with  relevant results cited in this survey would help contextualize the current contribution within the broader research landscape. For example, in results such as those by Merrill and Subharwal involving ${\rm CoT}(T(n))$, what is the implicit or explicit window size? Is it effectively $T(n)$? Clarifying how this relates to the ${\rm WINDOW}(s(n))$ definition would strengthen the positioning of the result.

---

> ### Author Rebuttal · Authors · 2025-07-29
>
> We sincerely thank the reviewer for the valuable comments and feedback. We provide a detailed clarification to address the reviewer’s concerns.
>
>
>
> **Q1:**  The literature review could be more thorough. The recent survey “What Formal Languages Can Transformers Express?” by Strobl et al. offers a thorough overview of expressivity results.....For example, in results such as those by Merrill and Subharwal involving , what is the implicit or explicit window size? Is it effectively ? Clarifying how this relates to the definition would strengthen the positioning of the result.
>
> **A1:** We will broaden the related work section as suggested. In particular, also suggested by Reviewer Y1kW, we will add a table comparing the existing Turing-completeness proofs in terms of required precision, embedding size, positional encoding scheme, effective window size, and CoT length.
>
> All existing Turing-completeness proofs, including that of Merrill and Subharwal (2023), require the context window to cover the entire context. So the implicit window length is $O(T(n)+n)$, where $T(n)$ is the number of CoT steps.
>
>
>
> **Q2:** It would be interesting for the authors to explore and discuss the implications of their results when is sublinear (e.g., . Does $\mathrm{WINDOW}(O(\log n))$ capture LOGSPACE? Are constant window transformers interesting?
>
> **A2:** Note that the entire input string is presented before the transformer starts generating CoT tokens, and at every step the transformer can look back only $s(n)$ positions.  If $s(n)<n$, the first $n−s(n)$ input symbols are permanently outside the transformer's attention and can never influence any CoT token.  Consequently, $\mathrm{WINDOW}(O(\log⁡n))$ does not capture LOGSPACE. In fact, it cannot recognize any language that depends on the whole input.

---

> > ### Comment · Reviewer_AC2j · 2025-08-04
> >
> > Thanks for the response. I am glad that you will add more related work and mitigate my concern. I like the contribution, so I will update my score to 5.

---

### Official Review · Reviewer_PMuH · 2025-07-02

**Clarity:** 2
**Significance:** 2
**Originality:** 2
**Rating:** 4
**Confidence:** 4

**Summary:**

This paper presents a theoretical result on the computational power of Transformers. The authors prove that a Transformer with a constant bit-size (i.e., fixed number of parameters and fixed numerical precision) is Turing complete, capable of simulating any Turing machine on inputs of arbitrary length, provided its context window is sufficiently long. Theoretical statements and proofs are provided.

**Questions:**

See the Weaknesses part above.

**Ethical Concerns:**

["NO or VERY MINOR ethics concerns only"]

**Final Justification:**

The rebuttal addressed my concerns, so I am raising my score to 4.

**Limitations:**

The authors do not discuss limitations of this work.

**Paper Formatting Concerns:**

The main body of this paper does not meet 9 pages.

**Quality:**

2

**Strengths And Weaknesses:**

## Strengths
- The paper is well-motivated and clearly written.
- The paper provides more than a simple "yes/no" answer to Turing completeness. The equivalence creates a direct and formal link between architectural parameters (e.g., context length) and computational budgets (memory/space).

## Weaknesses
- **Expressivity vs. Learnability**: The proof is a constructive one, meaning it shows that a Transformer with specific, carefully engineered parameters exists. It does not address whether this complex, algorithmic behavior could be learned using standard gradient-based training methods. This is a significant gap between the theoretical capability and practical reality.

- **Computational Cost of Simulation**: The simulation requires a number of chain-of-thought (CoT) steps that scales with O(t(n) * s(n)), where t(n) is the time complexity of the original Turing Machine. For many problems, this is prohibitively large and far exceeds the number of reasoning steps seen in practice, which may limit the direct applicability of the result.

- **Idealized Model Assumptions**: The construction relies on an idealized Transformer model that uses hardmax attention and omits layer normalization. While the authors note these are common simplifications in theoretical work, it creates a disconnect from the standard architectures used in practice.

- **Lack of Empirical Validation**: The work is purely theoretical. While the claims are powerful, the paper would be strengthened by even simple experiments that explore or demonstrate aspects of the proposed construction, which could provide insights into its feasibility or the challenges in realizing it.

---

> ### Author Rebuttal · Authors · 2025-07-29
>
> We appreciate the valuable comments and feedback. We provide a detailed clarification hoping to address the reviewer’s concerns.
>
>
>
> **Q1:** About Expressivity vs. Learnability.
>
> **A1:** Our contribution is purely about expressiveness. Whether standard training procedures can learn such behavior is outside our current scope and remains an important open problem. We will cite it as a future direction in the next version.
>
>
>
> **Q2:** About Computational Cost of Simulation.
>
> **A2:** We acknowledge this limitation, and have already cited it as an open problem in the Conclusion.
>
>
>
> **Q3:** About Idealized Model Assumptions.
>
> **A3:** Hardmax vs. softmax: Hardmax is a standard theoretical proxy, see e.g. [Perez et al. 2021], [Merrill and Sabharwal 2024], and [Li et al. 2024]. Importantly, our transformer construction is robust to switching to softmax: a softmax with any sufficiently small temperature will select the same unique-max token with probability $\approx1$.
>
> Layer normalization: As noted in our footnote,  our results can be extended to transformers with layer normalizations, by adopting the trick from [Li et al. 2024]. Specifically, we can replace each constant-precision real number with a constant-dimensional $\pm 1$-valued vector whose coordinates sum to zero, so that layer normalizations levels the computation unchanged.
>
> We will make both points clearly in the next version.
>
>
>
> **Q4:** About Lack of Empirical Validation.
>
> **A4:** We agree that empirical experiments would be valuable for justifying the practical feasibility. We will cite it as a future direction in the next version.
>
>
> **Q5:** The authors do not discuss limitations of this work.
>
> **A5:** In the next version, we will add more discussions on limitations and future directions:
>
> - CoT length: Our current simulation needs $O(t(n) s(n))$ CoT steps; reducing this to $O ⁣(t(n))$ remains an open problem.
> - Learnability and empirical validation: Our contribution is purely about expressiveness. Whether standard training procedures can learn such behavior is an important open problem. How well implementations behave in practice also remains to be investigated.
> - Positional encodings: Our construction employs an nonstandard relative positional encoding; extending the result to fixed absolute positional encodings or other standard relative positional encodings is open.

---

### Official Review · Reviewer_aGq8 · 2025-07-02

**Clarity:** 3
**Significance:** 4
**Originality:** 3
**Rating:** 5
**Confidence:** 4

**Summary:**

Transformers with Chain of Thoughts (CoTs) were initially proven by [Perez et al. 2021] to be Turing-complete. Essentially, this means that they are allowed to feed the output back into the input several times (that is, "think out loud") before outputting the final result of the computation. In subsequent years, researchers have revisited the result from [Perez et al. 2021] by simplifying the proof, or removing some unpractical assumptions in the model. For example, in the following paper Bhattamishra et al. simplified some of the assumptions in [Perez et al.] including replacing positional encodings with positional masking:

Satwik Bhattamishra, Arkil Patel, Navin Goyal: On the Computational Power of Transformers and Its Implications in Sequence Modeling. CoNLL 2020: 455-475

In this paper and in [Perez et al. 2021], finite but unbounded precision is assumed. More precisely, this means that computation is done over rational numbers, which amounts to potentially infinitely many decimal places (it is possible to actually restrict finitely many decimal places, but the number of required decimal places will grow in the size of the input). This result has been further refined in [Merrill and Sabharwal 2023] with an even simpler usage of positional masking and additionally complexity analysis.

A different set of assumptions for Turing-completeness was obtained by [Liu et al. 2024], where the authors use an unbounded "embedding dimension" but constant precision. Roughly speaking, this means that different parameter instantiations (possibly of different dimensions) exist for different input lengths. [Actually, the paper itself doesn't talk about another proof of Turing completeness of transformers, but as the current submission also acknowledges it follows from [Liu et al. 2024]].

The current paper improves on and simplifies the result by [Liu et al. 2024] in that the dependence on the input length exists only in the relative positional encoding; the other parameters stay the same. More precisely, the bit size of a transformer is essentially the number of bits (with only floating points allowed, not rational numbers) required to represent the parameters of the transformer. Crucially, a relative positional encoding is allowed as a parameter. With this assumption, the authors establish a "universal CoT transformer" with a constant bitsize. Actually, this translates to non-fixed (absolute) positional encoding that depends on the input length. Additionally, the current paper also assumes a sliding window of a non-fixed size on the concatenation of the input and the intermediate CoT results, which is reasonable.

Additionally, the paper also proves a characterization of the complexity class SPACE[s(n)] of problems solvable in s(n) space, by allowing a sliding window of size s(n). The proof goes by encoding a universal Post Machine (a queue machine) by means of a CoT transformer. Note that the number of bits required to store values in the intermediate computation might also grow in the length of the input (for example, owing to the use of average hard attention mechanism, which divides by a non-fixed number).

**Questions:**

- Can you say what happens in the case of a fixed "absolute" positional encoding?
- Is it possible to replace your use of positional encoding with a casual positional masking?
- Can you explain key technical differences in the proofs in your paper compared to [Perez et al. 2021] and [Li et al. 2024]?

**Ethical Concerns:**

["NO or VERY MINOR ethics concerns only"]

**Final Justification:**

The authors addressed my questions. I keep my accept score.

**Limitations:**

Yes

**Quality:**

4

**Strengths And Weaknesses:**

To the best of my knowledge, this is the first Turing-completeness proof that assumes some sort of constant bit size in the entire parameter set. The only catch is that one has to use relative positional encoding, which translates to a parametric absolute positional encoding. That said, unlike [Li et al. 2024], this is the *only* parametric component; the others are fixed. To the best of my knowledge, this is the strongest Turing-completeness result under the "constant precision" regime.

The proof is also relatively simple and quite nicely presented. Essentially, the queue content in the post machine corresponds to the sliding window content in the CoT transformers. The explicit construction also makes it clear what varies in the size of the input. My only quibble here is that the authors have 1.5 extra pages, which can be used to provide more intuition of the proof (as well as some definitions).

In addition, I found that some definitions are excessively overcomplicated. For example, multi-head attention is defined in the paper, whereas only single-head attention is used in the proof.

Overall, I think the strengths overweigh the weaknesses in the paper and I am leaning towards acceptance.

Minor:
- l. 145: "in some state"
- l. 180: $s^l_{k,i,j}$ is not defined. Do you mean the jth entry of $s^l_{k,i}$?
- l. 247: at first sight, it seems that you need to know i and j in pos(i-j), instead of just the difference i-j. Justify that this is not the case.

---

> ### Author Rebuttal · Authors · 2025-07-29
>
> We greatly value the reviewer's suggestions and recommendations. We will revise the related parts accordingly.
>
> **Q1:** Note that the number of bits required to store values in the intermediate computation might also grow in the length of the input (for example, owing to the use of average hard attention mechanism, which divides by a non-fixed number).
>
> **A1:** In our construction, the attention weight is always one-hot: at each step, exactly one token in the window receives a non-zero score. So the hardmax output is $\{0,1\}$-valued, no division ever occurs, and the required precision stays constant.
>
> As an alternative, one could replace hardmax with the right-most hardmax used in the paper “Masked Hard-Attention Transformers Recognize Exactly the Star-Free Languages” (We thank Reviewer Y1kW for this suggestion).
>
>
> **Q2:** Can you say what happens in the case of a fixed "absolute" positional encoding?
>
> **A2:** *With the constant bit-size constraint.* Whether a constant bit-size transformer with a fixed absolute positional encoding can still be Turing complete is an open problem. We currently have no proof either way.
>
> *Without the constant bit-size constraint.* Our second core result still holds: it remains sufficient for the context window length to scale with the space complexity of the simulated Turing machine.  A simple choice of the fixed absolute PE is
>
> $\mathrm{poi}(i)= i \mod p$ where $p>s(n)$.
>
> In our proof, the Post machine's queue size keeps exactly $s(n)$ throughout the computation, or equivalently, that the distance between the front and rear heads is fixed at $s(n)$. So in each CoT step, the query attends to the only token exactly $s(n)$ positions earlier, which can retrieved with this absolute PE alone.
>
>
>
> **Q3:** Is it possible to replace your use of positional encoding with a casual positional masking?
>
> **A3:** Yes.  As noted in A2, in our construction, the query attends to the only token exactly $s(n)$ positions earlier. We can therefore switch to a causal positional masking scheme that allows token $i$ to attend only to token $i−s(n)$. All other parts of the proof remain unchanged.
>
>
> **Q4:** Can you explain key technical differences in the proofs in your paper compared to [Perez et al. 2021] and [Li et al. 2024]?
>
> **A4:**
> - *[Perez et al. 2021] and related follow-ups.*  In these proof, the query may attend to many earlier tokens. When multiple positions tie for the maximal score, the hardmax outputs fractions such as $1/t$, which requires the precision to grow with $t$.
>
> - *[Li et al. 2024].* This proof encodes the simulated circuit directly into the absolute PE (rather than into the parameters). As the circuit (hence the input length) expands, the PE must grow in dimension, so the embedding size is no longer constant.
>
> - *This work.* The Post machine’s automaton-like behavior is much simpler and more regular than the head-moving  behavior of TMs and the wiring of a circuit. This let our construction satisfy a nice property: at every CoT step the query attends to exactly one token, always the token $s(n)$ positions earlier. This fixed-offset attention can be realized solely by a relative positional encoding without any additional arithmetic operations.
>
> In addition, we will also revised the other part as suggested, including adding the missing references, simplifying excessively overcomplicated definitions, and fixing typos.

---

> > ### Comment · Reviewer_aGq8 · 2025-08-04
> >
> > Thank you for the clarification. I maintain the accept score.

---

### Decision · Program_Chairs · 2025-09-17

**Decision:**

Accept (poster)

**Comment:**

**Summary of Claims and Findings**：
This paper presents a significant theoretical result on the computational power of Transformers. The authors prove that a Transformer with a constant bit-size—meaning a fixed number of parameters and fixed numerical precision—is Turing complete. The central finding establishes a precise equivalence: a constant bit-size Transformer with a context window of length $s(n)$ can recognize exactly the languages in the complexity class $SPACE[s(n)]$. The proof is achieved by simulating Post machines, a Turing-complete computational model whose queue-based operations align naturally with the autoregressive, windowed attention of Transformers.

**Strengths**:
* **Strong Theoretical Contribution:** This is the first work to establish Turing completeness for Transformers while maintaining a constant bit-size for all parameters and internal computation, improving upon prior results that required scaling precision or embedding dimensions with the input length. However, a similar idea is also explored in a preprint [Schuurmans et al.,24], though they do not focus on the constant precision. The authors should compare with [Schuurmans et al.,24] carefully in revision.
* **Precise Characterization:** The paper goes beyond a simple Turing-completeness claim by formally linking the Transformer's context window size ($s(n)$) to the space complexity ($SPACE[s(n)]$) of the problem, providing a tight characterization.
* **Clarity and Novel Proof Technique:** Reviewers consistently praised the paper for its clarity and the simple and clear proof structure. The use of Post machines provides a more intuitive and natural mapping to the Transformer architecture compared to direct Turing machine simulations.

**Weaknesses**:

The main issue of the paper, is that the cited theorem 2, which is the foundation of the entire paper, cannot be found in the provided reference. The closet result I am aware of in literature, are "Small Universal Circular Post Machines" by Manfred Kudlek and Yurii Rogozhin in 2001 and "Autoregressive Large Language Models are Computationally Universal" by Dale Schuurmans, Hanjun Dai, and Francesco Zanini in 2024. Still the results there are weaker than theorem 2 cited in this paper. For example, the Theorem 7 in Schuurmans et al. requires the transformer/post machine to read the leftmost **two** symbols and then decide what to write on the tape next. In contrast, Theorem 2 in this submission only requires reading the leftmost symbol only, which appears stronger to me.

Below are some other weakness:
* **Idealized Model Assumptions:** The construction relies on a non-standard relative positional encoding and uses hardmax instead of softmax attention, creating a gap with architectures used in practice.
* **Hidden Requirement of High Precision**: The main reason that this construction of transformer with window attention does not need log precision is because the high-precision number S(n) is hard-coded into the inference system. As reviewer Y1kW pointed out, it is not clear how transformer can figure that our automatically using constant precision.
* **High Simulation Cost:** The number of chain-of-thought (CoT) steps required for the simulation of Turing machine running in $t(n)$ steps and $s(n)$ space, $O(t(n)s(n))$, is potentially prohibitive for practical applications. In contrast, previous work  [Yang et al., 25] provide same simulation using only $O(t(n))$ steps CoT with a different space-saving mechanism.

**Reason for Recommendation**:

The paper makes an interesting and well-presented contribution to the theoretical understanding of Transformers. The primary reason for acceptance is significance of proving Turing completeness under the realistic constraint of a constant bit-size and the novelty as well as simplicity of the proof. The precise equivalence established between window size and space complexity is a powerful result that advances the field, assuming the correctness of Theorem 2. While there are the weaknesses regarding idealized assumptions and the expressivity-learnability gap, they are common limitations for purely theoretical work in this domain and do not detract from the paper's core contribution.

After weighing the strengths and weaknesses, the final recommendation for this paper is Accept(poster), conditioned on that the authors will provide a proper reference for their Theorem 2, or alternatively, a rigorous proof for it. This is the conclusion that SAC and AC have reached after a long discussion. We do expect the authors to withdraw the paper if they cannot do so.

#References:

- Kudlek, M. and Yu, R., 2001. Small universal circular Post machines. Comput. Sci. J. Mold, 9, p.25.

- Yang, C., Srebro, N., McAllester, D. and Li, Z., PENCIL: Long Thoughts with Short Memory. In Forty-second International Conference on Machine Learning.

- Schuurmans, D., Dai, H. and Zanini, F., 2024. Autoregressive large language models are computationally universal. arXiv preprint arXiv:2410.03170.